# Association between muscle strength and depression in a cohort of young adults

**Tomáš Vodička**[1]*, **Michal Bozděch**[2], **Tomáš Vespalec**[1], **Pavel Piler**[3], **Ana Carolina Paludo**[4]

1 Department of Physical Activities and Health Sciences, Faculty of Sports Studies, Masaryk University, Brno, Czech Republic, 2 Department of Physical Education and Social Sciences, Faculty of Sports Studies, Masaryk University, Brno, Czech Republic, 3 Faculty of Science, RECETOX, Masaryk University, Brno, Czech Republic, 4 Department of Sport Performance and Exercise Testing, Faculty of Sports Studies, Masaryk University, Brno, Czech Republic

* tvodicka@fsps.muni.cz

**Data Availability Statement:** All relevant aggregated data are within the manuscript. Anonymized data from each study participant can be available upon request. Release of data is a

## Abstract

### Background

The study investigated the association between knee joint muscle strength and the prevalence of depression in a cohort of young adults.

### Methods

The observational, population-based study was performed with 909 participants (29.02 ± 2.03 years; 48.73% male) from the Central European Longitudinal Studies of Parents and Children: Young Adults (CELSPAC: YA), who were retained to analysis. Quadriceps and hamstring knee muscle strength were assessed by isokinetic dynamometry, and depression by Beck's Depression Inventory (BDI-II). Statistical comparisons (Mann-Whitney and Chi-squared test) and effect size analyses (Eta-Squared, and Odds Ratio) were conducted.

### Results

The main findings revealed an inverse association between knee joint muscle strength and depression, with individuals who had low muscle strength having 3.15 (95% CI = 2.74–3.62) times higher odds of experiencing depression. Specifically, participants with low extensor strength had 4.63 (95% CI = 2.20–9.74) times higher odds, and those with low flexor strength had 2.68 (95% CI = 1.47–4.89) times higher odds of experiencing depression compared to those individuals with high muscle strength. Furthermore, gender-specific analyses revealed that males with low muscle strength had 2.51 (95% CI = 1.53–4.14) times higher odds, while females had 3.46 (95% CI = 2.93–4.08) times higher odds of experiencing depression compared to individuals with high muscle strength.

### Conclusions

Strong knee muscles seems to be a key factor in preventing depression, specially in female young adults. The results support the importance of promoting an increase in muscle

subject of approval of the CELSPAC Ethical and Scientific Board. The contact is here: info@celspac.cz.

**Funding:** The author(s) received no specific funding for this work.

**Competing interests:** The authors have declared that no competing interests exist.

strength through physical activity as a preventive strategy against depression in this population.

## Introduction

The positive impact of high levels of physical activity on depression in the adult population is well-documented [1]. Consequently, due to physical activity being a modifiable factor related to muscular strength, recent studies have focused on evaluating the connection between muscle strength and the prevalence of depression [2].

Depression is one of the most prevalent and personally debilitating mental health disorders, posing a significant public health issue in contemporary times. Nowadays, it affects more than 280 million people, and its prevalence is still significantly increasing [3]. Inseparably liked with poor health [4], including an increased risk of cardiovascular diseases [4] and type 2 diabetes [5], as well as being a leading cause of suicide [6].

Within the workforce, depression stands out as a substantial contributor to absenteeism and disability [7]. Furthermore, the total costs of mental health problems are estimated to be more than 4% of the total gross domestic product (more than EUR 600 billion) across the 27 European countries and the United Kingdom [8]. The prevalence of depression is nearly twice as high in females compared to males across all ages, and both genders experience a peak in prevalence during their second and third decades of life [9, 10]. Currently, medication and psychotherapy, are the main treatments for depression. However, drug treatments are hindered by side effects, addiction, high prices, and poor patient compliance, resulting in an overall unsatisfactory and seriously affected quality of life for patients. [11]. Moreover, psychotherapy can be expensive and inaccessible, and its overall effects can be overestimated [12]. Given the breadth of depressive disorders, strategies that may reduce the onset of depression are urgently needed.

Exercise interventions demonstrate promise as viable treatments for depressive symptoms, presenting benefits in comparison to antidepressant medications and psychotherapy due to their minimal adverse effects and reduced expenses. Furthermore, current studies are focused on how muscle strength can be perceived as a modifiable factor related to lower levels of depression prevalence. Recently, a 7-year follow-up study with 5,228 participants demonstrated that higher relative handgrip strength was a protective factor against depression in the adult population [13]. While the handgrip strength test has commonly been utilized as a parameter of muscle strength in populational studies due to its non-invasiveness, low cost, and practicality, it is worth noting that the activation of muscle groups during this test is limited and may have lower applicability when considering daily physical activities. Therefore, it seems reasonable to investigate muscle strength by focusing on major muscle groups.

Methods such as walking pace [14, 15] and the Sit to Stand Test [13] have been recently used in examining the association between lower limb strength and depression, particularly in the elderly population. Considering that the young adult population is more sensitive to depression [16], it is necessary to investigate the possible association between depression and low limb muscle strength using a gold-standard method.

Therefore, the main aim of the study is to examine whether there is an association between high levels of low limb muscle strength and depression in the young adult population. For this purpose, the study will use the gold-standard method for muscle strength testing, isokinetic dynamometry, to measure the strength of the knee extensor and flexor muscles. The study will also investigate associations between depression and knee joint muscle strength in both males

and females. We hypothesized that a higher level of muscular strength in the lower limbs will be associated with lower scores with depressive symptoms, with a stronger association in females compared to males.

## Materials and methods

### Study design, setting and sample

This observational population-based study originates from the Central European Longitudinal Studies of Parents and Children: Young Adults (CELSPAC: YA). Young Adults cohort is an on-going follow-up study of the Czech part of the ELSPAC birth cohort (European Longitudinal Study of Pregnancy and Childhood) that was initiated in 1991–1992 in the Czech Republic, in Brno and Znojmo region. Detailed information about the ELSPAC-CZ study is provided elsewhere [17]. For the current research goal, data from participants in the CELSPAC: YA cohort were collected between March 1, 2019, and February 1, 2023. The data were accessed for research purposes on February 15, 2023. Authors had access to information that could identify individual participants after data collection. The CELSPAC: YA study was approved by the ELSPAC Ethics Committee (Ref. No: ELSPAC/EK/2/2019), and all participants of this study provided written informed consent.

To examine the relationship between muscle strength and the incidence of depression, the variables were selected by using isokinetic muscle strength testing and questionnaires. Questionnaires including both the health condition and the health history were filled by the participants with health practitioner assistance. The depressive symptoms assessment and alcohol compulsion were filled out by the participants themselves. Participants who did not provide information regarding these variables were excluded. The study included participants who took part in the surveys and were evaluated for isokinetic knee muscle strength.

The following exclusion criteria were applied: participants with physical disabilities (such as chronic lower extremity pain, acute injuries, or injuries related to the knee joint) that could affect muscle strength measurements were not included in the study. Additionally, participants with previously diagnosed psychiatric disorders including schizophrenia, bipolar disorder, or substance abuse were also excluded from the analysis. From a total of 967 participants, 909 young adult participants meet the inclusion criteria and were retained and analyzed in this study. The participants descriptive statistics are presented in Table 1. A priori Power analysis for the Proportion test was conducted utilizing G*Power (3.1.9.6) software. For total of 909 participants the Power (1-b) exceeds 0.91 (with medium effect, α of 0.05 and allocation ratio of 0.957).

### Muscle strength measurement

To assess knee muscle strength, a calibrated isokinetic dynamometer (Humac Norm, Computer Sports Medicine, Inc., Stoughton, MA, USA) was used. Recent research by [18] has

**Table 1. Basic descriptive statistics of the study participants.**

| Variable | Age (yrs) | Weight (kg) | Height (cm) | BMI (kg/m$^2$) |
|---|---|---|---|---|
| Total | 29.02 (2.03) | 74.65 (16.02) | 174.94 (9.62) | 24.27 (4.21) |
| Sex | | | | |
| • Male | 29.17 (2.01) | 83.56 (14.48) | 182.12 (6.72) | 25.17 (4.02) |
| • Female | 28.87 (2.04) | 66.18 (12.41) | 168.11 (6.48) | 23.42 (4.21) |

BMI, Body Mass Index.

demonstrated the excellent reliability of the Humac Norm isokinetic dynamometer for testing knee joint muscle strength. For the evaluation of muscle strength, we conducted a similar testing protocol as described in the study by [19]. Briefly, participants were seated in the isokinetic dynamometer chair with the back support set at an angle of 85˚. The pad of the dynamometer was positioned approximately 3 cm above the lateral malleolus. The knee joint axis was carefully aligned with the mechanical axis of the dynamometer. The testing protocol began by evaluating the dominant limb first. To warm up and familiarize themselves with the movements, participants performed five non-maximal trials on the dynamometer for each movement. Following a thirty-second pause, the concentric isokinetic knee flexion and extension movements were assessed at an angular velocity of 60 degrees per second ($60˚/s^{-1}$). Each movement consisted of five maximal repetitions over a range of motion of 90 degrees, from 0˚ (full knee extension) to 90˚ of knee flexion. The maximal knee extensor and flexor strengths were assessed by measuring the peak torque (in Newton meters, Nm) during the isokinetic concentric contraction. During the tests, the investigator provided verbal encouragement to help participants achieve their maximal strength. Participants were not permitted to view the screen during testing. Prior to each test, gravity correction was obtained to ensure accurate measurements.

## Depression assessment

The severity of depression was assessed using the second edition of the Beck Depression Inventory (BDI-II) questionnaire. The validity and reliability of the BDI-II for screening of depression is well established [20]. The BDI-II is a self-report questionnaire that assesses symptoms of depression and has a strong correlation with clinical diagnosis of depression [21]. The BDI-II consists of 21 items, and participants rate each item on a Likert scale ranging from 0 to 3. Higher scores on the questionnaire indicate more severe depressive symptoms. Total scores on the BDI-II questionnaire can range from 0 to 63. The classification of depression severity is as follows: scores between 0 and 13 are classified as no depression, scores between 14 and 19 are classified as mild depression, scores between 20 and 28 are classified as moderate depression, and scores between 29 and 63 are classified as severe depression [22]. For this study, the BDI-II questionnaire has been translated and standardized into the Czech language.

## Sociodemographic and lifestyle characteristics

Supplementary variables were determined at baseline. The sociodemographic characteristics included age, sex, height, and weight. Lifestyle factors included alcohol consumption. The anthropometric characteristics of participants were measured using a digital scale, (Seca 285, Hamburg, Germany). Standing height (cm) and weight (kg) were measured. The body mass index body mass index (BMI; weight/height$^2$) was used to classify participants as underweight ($<18.5 \, kg/m^2$), normal weight ($18.5–24.9 \, kg/m^2$), overweight ($25.0–29.9 \, kg/m^2$), and obese ($\geq 30 \, kg/m^2$).

The lifestyle about participants' alcohol consumption habits were self-reported using survey questionnaires. Alcohol consumption was classified as follows: never (never to 3 times a month), moderate (1–4 times a week), and heavy (5–7 times a week).

## Statistical analysis

A Mann-Whitney $U$ test was used for continuous data, while a Chi-squared test of independence ($\chi^2$) was employed for categorical data in order to assess differences between the researched groups in terms of baseline characteristics. An effect size test was also conducted for both tests, specifically using Cramer's V (V, $df_{min} = 1$) for the Chi-squared test and Eta-

Squared (h$^2$) for the Mann-Whitney U test. The effect size results were interpreted as small (V = 0.10; η$^2$ = 0.01), medium (V = 0.30; η$^2$ = 0.06) or large (V = 0.50; η$^2$ = 0.14) effect [23]. The distribution of participants with and without depression, as well as the tertiles (T1 –T3) of relative muscular strength (Nm/Kg), was calculated using Chi-square goodness of fit test, assuming equal expected frequencies [24]. The adjustment estimation for the random-effects model utilized the log Odds Ratio (OR) for binary outcomes in a 2 by 2 table to quantify the odds of participants with lower muscle strength having depression compared to participants without depression. To determine if the effect size was consistent across and between the investigated variables (movements), a test of homogeneity (Q statistic) was performed. The significance level was set at $p$ = .05. The analysis was conducted using IBM SPSS Statistics for Windows version 29.0.0 software (IBM Corp. Armonk, NY, USA).

## Results

A total of 909 participants were included in the study, with an average age of 29.02 ± 2.03 years and 48.73% of them being male. Table 1 provides additional details on the anthropometric characteristics of the participants. Table 2 presents the characteristics of the participants and the prevalence of depression. Individuals with depression were more likely to be female ($p$ = .031) and have normal weight ($p$ = .045) in comparation to individuals without depression.

The relative isokinetic muscle strength of various muscle groups of the knee joint (n = 8) was separated into tertiles (T1 –T3) based on z-scores. T1 corresponded to low muscle strength (z-score < 1), T2 corresponded to average muscle strength (z-score ± 1), and T3 corresponded to high muscle strength (z-score > 1). In order to focus on extreme outcomes (T1 and T3), participants with average muscle strength (T2) were excluded from the study. The low muscular strength group consisted of 131 to 149 participants, while the high muscular strength group consisted of 127 to 152 participants, as shown in Table 3.

Table 4 presents the results observed in all participants using dichotomized outcomes for muscle strength (low vs. high) and depression (no depression vs. depression). An inverse

**Table 2. Participants characteristics and depression prevalence.**

| Variable | Depressed (*n* = 150/16.50%) | Not depressed (*n* = 759/83.50%) | Total (*n* = 909/100%) | *P* | ES |
|---|---|---|---|---|---|
| Weight, kg | 75.26±17.44) | 74.53±15.74 | 909 | .816 | <0.001 |
| Height, cm | 174.06±9.08 | 175.11±9.73 | 909 | .243 | 0.001 |
| Sex, *n* (%) | | | | | |
| • Male | 61 (40.67) | 382 (50.33) | 443 (48.73) | .031 | 0.072 |
| • Female | 89 (59.33) | 377 (49.67) | 466 (51.27) | | |
| BMI, *n* (%) | | | | | |
| • Underweight | 9 (6.00) | 24 (3.16) | 33 (3.63) | .045 | 0.094 |
| • Normal Weight | 80 (53.33) | 474 (62.45) | 554 (60.95) | | |
| • Overweight | 42 (28.00) | 202 (26.61) | 244 (26.84) | | |
| • Obese | 19 (12.67) | 59 (7.77) | 78 (8.58) | | |
| Alcohol intake, *n* (%) | | | | | |
| • Never | 56 (37.33) | 285 (37.55) | 341 (37.51) | .988 | 0.005 |
| • Moderate | 83 (55.33) | 421 (55.47) | 504 (55.45) | | |
| • Heavy | 11 (7.33) | 53 (6.98) | 64 (7.04) | | |

$p$-value was calculated using Mann-Whitney U test for continuous data and Chi-squared test of Independence (χ$^2$) for categorical data; ES χ$^2$ for continuous data and Cramer's V for categorical data.

**Table 3. Participants relative muscle strength according to z-score.**

| Variable | n | Relative muscular strength (Nm/kg) | | | |
|---|---|---|---|---|---|
| | | T1 | T2 | T3 | p |
| | | Low (z< 1SD) | Average (z±1SD) | High (z>1SD) | |
| EXT_R | 908 | 149 (16.59%) | 622 (69.27%) | 137 (14.14%) | < .001 |
| EXT_L | 909 | 131 (14.41%) | 651 (71.62%) | 127 (13.97%) | < .001 |
| FLEX_L | 909 | 148 (16.28%) | 609 (67.00%) | 152 (16.72%) | < .001 |
| FLEX_R | 908 | 141 (15.53%) | 617 (67.95%) | 150 (16.52%) | < .001 |
| FLEX+EXT_L | 909 | 132 (14.52%) | 645 (70.96%) | 132 (14.52%) | < .001 |
| FLEX+EXT_R | 908 | 135 (14.87%) | 638 (70.26%) | 135 (14.87%) | < .001 |
| EXT_R+L | 909 | 139 (15.29%) | 637 (70.08%) | 133 (14.63%) | < .001 |
| FLEX_R+L | 909 | 145 (15.95%) | 616 (67.77%) | 148 (16.28%) | < .001 |

EXT_R, muscle strength right knee extensors. FLEX_L, muscle strength left knee flexors. FLEX_R, muscle strength right knee flexors. EXT_L, muscle strength left knee extensors. FLEX+EXT_L, muscle strength thigh muscles left leg. FLEX+EXT_R, muscle strength thigh muscles right leg. EXT_R+L, muscle strength right and left extensors. FLEX_R+L, muscle strength right and left flexors. p-value was calculated using chi-square goodness of fit test with equal expected distribution (33.3%).

association between the muscle strength of the knee joint and depression were found, demonstrating that individuals with low muscle strength of the knee joint having 3.15 times higher odds of having depression (95% CI = 2.74–3.62) compared to those with high muscle strength. Specifically, participants with low extensor strength had 4.63 (95% CI = 2.20–9.74) times higher odds, while those with low flexor strength had 2.68 (95% CI = 1.47–4.89) times higher odds of experiencing depression when compared to individuals with high muscle strength. Considering limb preference, the odds of experiencing depression in participants with low muscle strength were more pronounced in the extensors and flexors of the right limb (OR = 3.64, 95% CI = 1.87–7.08) compared to the left limb (OR = 2.92, 95% CI = 1.51–5.66), when compared to individuals with high muscle strength. Additional information about descriptive statistics from muscle strength according to tertiles are presented in (S1 Table).

Due to evidence of interactions between depression and sex, separate analyses were conducted for males and females. In brief, knee muscle strength was found to be inversely associated with depression in both genders, except for knee flexors in males. Males with low muscle strength have 2.51 (95% CI = 1.53–4.14) times higher odds of experiencing depression compared to males with high muscle strength. Females with low muscle strength had 3.46 (95%

**Table 4. Overview of effect size results from participants muscle groups and depression.**

| Variable | OR | 95% CI | p | Weight | Weight (%) |
|---|---|---|---|---|---|
| EXT_R | 3.01 | [1.55, 5.84] | .001 | 8.70 | 2.85 |
| FLEX_L | 2.76 | [1.50, 5.08] | .001 | 10.28 | 3.37 |
| FLEX_R | 3.09 | [1.63, 5.85] | < .001 | 9.44 | 3.09 |
| EXT_L | 3.30 | [1.69, 6.45] | < .001 | 8.52 | 2.79 |
| FLEX+EXT_L | 2.92 | [1.51, 5.66] | .001 | 8.82 | 2.89 |
| FLEX+EXT_R | 3.64 | [1.87, 7.08] | < .001 | 8.68 | 2.84 |
| EXT_R+L | 4.63 | [2.20, 9.74] | < .001 | 6.93 | 2.27 |
| FLEX_R+L | 2.68 | [1.47, 4.89] | .001 | 10.59 | 3.47 |
| Overall effect | 3.15 | [2.74, 3.62] | .001 | | |

p-value was calculated using Odds ratio test (OR); CI, confidence interval.

**Table 5. Overview of effect size results from muscle strength and depression according to sex.**

| Subgroup | Variable | OR | 95% CI | p | Weight | Weight (%) |
|---|---|---|---|---|---|---|
| Male | EXT_R | 4.16 | [1.28, 13.47] | .018 | 2.80 | 0.9 |
| | FLEX_L | 1.63 | [0.61, 4.37] | .333 | 3.94 | 1.30 |
| | FLEX_R | 0.89 | [0.28, 2.84] | .846 | 2.87 | 0.90 |
| | EXT_L | 4.45 | [1.52, 13.05] | .006 | 3.32 | 1.10 |
| | FLEX+EXT_L | 3.79 | [1.29, 11.12] | .015 | 3.31 | 1.10 |
| | FLEX+EXT_R | 3.93 | [1.19, 13.01] | .025 | 2.69 | 0.90 |
| | EXT_R+L | 3.21 | [1.15, 8.99] | .026 | 3.63 | 1.20 |
| | FLEX_R+L | 1.44 | [1.53, 3.91] | .476 | 3.84 | 1.30 |
| Subgroup overall | | 2.51 | [1.53, 4.14] | .001 | | |
| Female | EXT_R | 3.06 | [1.28, 7.32] | .012 | 5.05 | 1.70 |
| | FLEX_L | 3.05 | [1.13, 8.23] | .028 | 3.9 | 1.30 |
| | FLEX_R | 2.85 | [1.25, 6.50] | .013 | 5.67 | 1.90 |
| | EXT_L | 3.18 | [1.29, 7.84] | .012 | 4.73 | 1.50 |
| | FLEX+EXT_L | 3.12 | [1.22, 7.95] | .017 | 4.38 | 1.40 |
| | FLEX+EXT_R | 5.00 | [1.98, 12.65] | < .001 | 4.47 | 1.50 |
| | EXT_R+L | 3.89 | [1.53, 9.92] | .004 | 4.39 | 1.40 |
| | FLEX_R+L | 4.25 | [1.61, 11.21] | .003 | 4.09 | 1.30 |
| Subgroup overall | | 3.46 | [2.93, 4.08] | .001 | | |

p-value was calculated using Odds ratio test (OR); CI, confidence interval.

CI = 2.93–4.08) times higher odds of experiencing depression compared to females with high muscle strength, as shown in Table 5 and Fig 1.

## Discussion

The study aimed to investigate the association between isokinetic muscle strength of the knee joint and depression. The results confirmed the hypothesis, indicating that young adults with low muscle strength were at 3.15 times higher odds of experiencing depression compared to those with high muscle strength. Additionally, the study found that individuals with a low level of extensor strength had 4.63 times higher odds of experiencing depression, while those with a low level of flexor strength had 2.68 times higher odds of depression compared to individuals with a high level of muscle strength. Furthermore, the study revealed that the isokinetic muscle strength of the knee joint was inversely associated with depressive symptoms in both sexes, however, a higher prevalence of depression was found in females. Specifically, females with low muscle strength had 3.46 times higher odds of experiencing depression compared to females with high muscle strength. Similarly, males with low muscle strength had 2.51 times higher odds of depression compared to males with high knee muscle strength.

The findings of our study align with previous research that has reported a causal relationship between depressive symptoms and low muscle strength in the lower limbs. For example, the study by [13] that utilized the five-repetitions sit-to-stand test (FRSTST) to examine the incidence of depression disorders during a seven-year follow-up in middle-aged and older adults, identified a hazard ratio of 1.32 (95% CI = 1.08–1.62) for the lowest quartile compared to the highest quartile of muscle strength (p = 0.007). In a recent meta-analysis, [25], concluded that adults aged 44 to 74 years who demonstrated slow gait speed had a pooled OR from 11 studies of 1.93 (95% CI = 1.54–2.42). Similarly, [15] revealed that senior participants

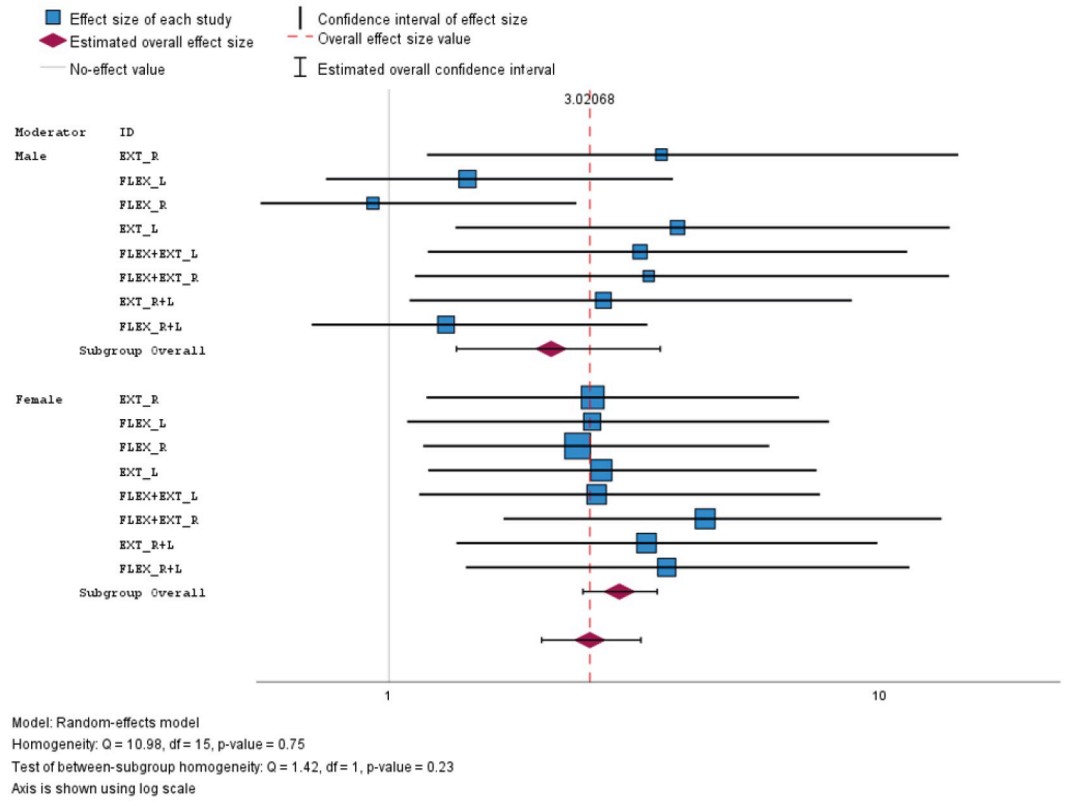

**Fig 1. Overview of effect size results according to sex.**

who performed significantly poorer in physical performance tests such as the 4-meter walking speed test, FRSTST, isometric leg strength, handgrip strength, and 6-minute walk test exhibited an elevated risk of developing depression over the 4-year follow-up period.

Besides the association between low muscle strength and risk of depression were similar with the aforementioned articles, it is possible to noticed that the young adults in the present study presented a higher odd (OR = 3.15) compared to studies with older adults and predictive tests (OR = 1.32–1.93). It can be speculated that either the young adult population present a higher odds of low muscle strength and risk of depression compared to older population, and also that the indirect measures can underestimate the results compared to the direct measure of low limb muscle strength. It is worth noting that previous studies often relied on predictive methods to assess lower limb muscle strength, whereas our study utilized a gold-standard method. Nonetheless, the use of predictive methods can increase the error by underestimating or overestimating the actual result. This strengthens the validity of our findings and provides more robust evidence for the association between depressive symptoms and lower limb muscle strength.

Regarding the different lower limb muscle strengths assessed, our study presented a stronger inverse association between muscle strength and depression in the extensor muscles compared to the flexor muscles. These findings reported that participants with low extensor muscle strength of the right and left limbs have 4.63 (95% CI = 2.20–9.74) higher odds of developing depression compared to participants with high muscle strength. It can be explained by the fact that knee extensor muscles are more involved in walking, and could be supported

by previous results that used walking tests and FRSTST which are widely recognized as indirect tests for assessing lower limb muscle strength, particularly knee extensor muscles [26, 27].

As expected, our study confirmed that young adult females exhibited a higher prevalence of depression compared to males. These findings align with previous research evidence, which also indicated a greater prevalence of depression among females than in males [28], and also with the WHO report, which estimated the prevalence of depression in the population to be higher among females (5.1%) than among males (3.6%) [10]. The increased prevalence of depression among females may be associated with hormonal changes.

Additionally, our study also revealed that participants with depression have a healthy weight compared to those without depression, which corresponds with the findings of [29], who also found that individuals with depression have lower BMI values. This association can be attributed to the fact that participants with normal weight have had the highest representation in our research group.

The physiological mechanisms related to higher level of muscle strength and anti-depressant effects remain hypothetical, despite its robust clinical effect. Several physiological theories have been proposed to clarify the anti-depressive effects of exercise affecting muscle strength, but the serotonin theory may be of particular significance.

The relationship between strength training and increasing of serotonin level can be explained by theory of [30], who suggested that strength training can increase the amount of free fatty acids and thus increase the levels of free tryptophan (TRP) in the bloodstream, influencing the availability and synthesis of serotonin (5-HT) in the Central Nervous System (CNS). It has been highlighted that any increase in the peripheral supply of TRP to the brain leads to an increased synthesis of 5-HT [31], thus the decreased peripheral 5-HT levels may reflect an increased transport of TRP into the CNS that would ultimately result in increased central 5-HT levels [32]. Furthermore, in the CNS, 5-HT modulates a broad spectrum of functions, including mood, cognition, anxiety, learning, memory, reward processing, and sleep [33], which can be associated with antidepressant effect.

Consistent with previous research, we have demonstrated that higher levels of muscle strength exert a beneficial effect on depression. Our study is the first to report the relationship between depressive disorders and knee muscle strength by using the gold-standard method.

The limitations of the study must be mentioned. Firstly, the nature of this study does not allow us to establish causal influences, highlighting the need for future well-designed longitudinal studies to clarify causality. Secondly, it is important to acknowledge that data collection for our study was conducted during the Covid-19 pandemic, which could have potentially influenced participants' perception of depression and led to reduced strength due to restrictions on physical activity. Additionally, our study did not focus on the influence of lifestyle characteristics, such as marital status, education level, and multi-comorbidity, on depression related to participants' age, as these characteristics may be incomplete. We also did not report the smoking status variables due to the various possible mechanisms of nicotine intake, such as cigarette smoking, e-cigarette use, vaping, waterpipe use, and smokeless tobacco products, where the nicotine dosage can vary. The influence of various types of nicotine intake on depression was reported elsewhere [34]. Regarding the sample, it is important to note that our study focused exclusively on young adult participants. As a result, caution should be exercised when generalizing the results to the broader population.

Despite the mentioned limitations, we firmly believe that our study holds significant implications for both research and clinical practice. Specifically, the increase in muscle strength through regular physical activity can be utilized as an effective tool to prevent and treat depressive disorders in the young adults' population. Furthermore, we highly recommend the aerobic and anaerobic interventions activities for both the prevention and additional treatment of

depressive disorders. Further research involving different populations is warranted in future studies to enhance the external validity of our findings.

## Conclusions

The study demonstrated that strong knee muscles seems to be a key factor in preventing depression, especially in Czech female young adults. To the best of our knowledge, this is the first population-based study that investigates the associations between knee joint muscle strength, assessed using the gold-standard method, and depressive symptoms in the young adult population. The current findings demonstrate that participants with low knee muscle strength have three times more chance to have depression compared to those with high muscle strength. Regarding the muscle groups, we found almost twice stronger inverse association between muscle strength and depression in the extensor muscles than in flexor muscles. Additionally, we observed a higher association between low muscle strength and depression in females than in males.

## Supporting information

**S1 Table. Overview of descriptive statistics from muscle strength according to tertiles.** (DOCX)

## Acknowledgments

We thank all collaborating participants who invested their time and provided information for this study.

## Author Contributions

**Conceptualization:** Tomáš Vodička, Tomáš Vespalec.

**Data curation:** Michal Bozděch.

**Supervision:** Ana Carolina Paludo.

**Writing – original draft:** Tomáš Vodička, Pavel Piler.

**Writing – review & editing:** Ana Carolina Paludo.

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
