## [Decision Letter · Decision Letter 0]

22 Jan 2024

PONE-D-24-00268Association between muscle strength and depression in a cohort of young adultsPLOS ONE

Dear Dr. Vodička,

Thank you for submitting your manuscript to PLOS ONE. After careful consideration, we feel that it has merit but does not fully meet PLOS ONE’s publication criteria as it currently stands. Therefore, we invite you to submit a revised version of the manuscript that addresses the points raised during the review process.

We look forward to receiving your revised manuscript.

Kind regards,

Julio Alejandro Henriques Castro da Costa

Academic Editor

PLOS ONE

Journal Requirements:

4. Please be informed that funding information should not appear in the Acknowledgments section or other areas of your manuscript. We will only publish funding information present in the Funding Statement section of the online submission form. Please remove any funding-related text from the manuscript.

Reviewers' comments:

Reviewer's Responses to Questions

**Comments to the Author**

1. Is the manuscript technically sound, and do the data support the conclusions?

Reviewer #1: Yes

Reviewer #2: Yes

2. Has the statistical analysis been performed appropriately and rigorously? 

Reviewer #1: Yes

Reviewer #2: Yes

3. Have the authors made all data underlying the findings in their manuscript fully available?

Reviewer #1: Yes

Reviewer #2: Yes

4. Is the manuscript presented in an intelligible fashion and written in standard English?

Reviewer #1: Yes

Reviewer #2: Yes

5. Review Comments to the Author

Reviewer #1: Abstract:

The methodology part needs more details, The Type of study design not clear.

The keywords should be more and according to the MeSh

Introduction:

The literature review needs to improve

The study's rationale needs to be explained more, why this study?

May you mention the fall risk in the significance of study and discussion

Methodology

The number of participants and The sample size calculation need to be explained to the participants

The isokinetic dynamometer procedure needs to have references 10.1177/1941738120986803

Add validity and reliability of the questionnaire

May you report the H/Q ration

Discussion

The discussion can improve and talk more about the fall risk

Need to be more critical in comparing your findings with previous studies

Add limitation of the study

Add strength and practical implications of the study

Reviewer #2: Thank you for the opportunity to review your study. The study aimed to examine whether there is an association between high levels of lower limb muscle strength and depression in a young adult population. The study contributes to the knowledge about the association between muscle strength and depression.

Minor comments

Please verify whether the authors mean gender (female and male) or sex (women and men). Include the study's definition of sex or gender, ensuring the appropriate use of the term.

Were the medication use, depression duration, physical activity level, and physical modality obtained from the participants? Were the participants diagnosed with depression before the study?

Was the statistical analysis adjusted for weight?

Lines 172 and 173: Did the participants with depression have a healthy weight or normal BMI?

Lines 274-276. Please add a hypothesis for the association between normal BMI and depression.

Lines 277-287: Please include the meaning of the 5-HT acronym. Include the explanation related to the role of serotonin on mental health status.

6. PLOS authors have the option to publish the peer review history of their article (what does this mean?). If published, this will include your full peer review and any attached files.

Reviewer #1: No

Reviewer #2: No

---

## [Author Response · Author response to Decision Letter 0]

20 Mar 2024

Dear Reviewer 1

We thank the reviewer for your evaluation and helpful comments on our manuscript. We have carefully taken your comments into consideration in preparing our revision, which has resulted in a paper that is clearer, broader and more compelling. Please find below our point-by-point responses to your comments, marked in the text with Track Changes.

Abstract:

1. The methodology part needs more details, The Type of study design not clear.

Reply: Thank you for the suggestion. In the methodological section of the abstract, we included the type of study design. Additionally, we provided details concerning the statistical analysis performed.

2. The keywords should be more and according to the MeSh

Reply: I appreciate the suggestion. We have revised and added keywords according with MeSH.

Introduction:

3. The literature review needs to improve 

Reply: The authors rewrote the introduction in order to improve. We have significantly revised the introduction section to enhance the literature review, incorporating additional references and providing a more comprehensive analysis of prior research. Please, if you have any additional suggestion or question, specify so we can identify the issue and solve it.

4. The study's rationale needs to be explained more, why this study?

Reply: Thank you for your constructive feedback. We have carefully considered your suggestion and have revised the manuscript accordingly to provide a more thorough explanation of the study's rationale.

5. May you mention the fall risk in the significance of study and discussion

Reply: Thank you for the comment. We believe that our study provides extensive information about the relationship between muscle strength of various muscles in connection with the prevalence of depressive disorders in the young adult population. We also believe that the fall risk is in this population is considerably small compared to other. In case of further study regarding older adults, we will definitely report the fall risk in connection with muscle strength and depression.

Methodology:

6. The number of participants and The sample size calculation need to be explained to the participants.

Reply: Thank you for your update. The authors incorporated the participant numbers and details about sample size calculation into the methods section.

7. The isokinetic dynamometer procedure needs to have references 10.1177/1941738120986803

Reply: The reference of the isokinetic dynamometer procedure was added as suggested.

8. Add validity and reliability of the questionnaire

Reply: Thank you for your comment. We have included information regarding the validity and reliability of the BDI-II questionnaire in the methods section.

9. May you report the H/Q ration

Reply: Thank you for the comment. Indeed. the reporting of the H/Q ratio is an interesting information, however in our study, it were mainly focused on providing detailed information about the strength profile of the extensors and flexors of the right and left extremities in connection with depression. The authors consider including this aspect in our future research with a different population sample.

Discussion:

10. The discussion can improve and talk more about the fall risk

Reply: Please see our response to your recommendation in the Introduction section regarding reporting the fall risk. Briefly, we believe that the fall risk is in young adult population negligible, we will report this in our future research in elderly population.

11. Need to be more critical in comparing your findings with previous studies

Reply: Indeed, a critical and deep comparison with previous studies is needed, however due to the different methodologies and population investigated in our study and the previous studies, the comparison can be compromised and biased.

12. Add limitation of the study 

Reply: In our study, we have already acknowledged the limitations of our research. However, following careful critical analysis, we have decided to rewrite this section to enhance clarity for the readers.

13. Add strength and practical implications of the study

Reply: Thank you for your comment. We have incorporated the strengths and practical implications of the study as suggested.

 

Dear Reviewer 2 

We thank the reviewer for your evaluation and helpful comments on our manuscript. We have carefully taken your comments into consideration in preparing our revision, which has resulted in a paper that is clearer, broader and more compelling. Please find below our point-by-point responses to your comments, marked in the text with Track Changes.

1. Please verify whether the authors mean gender (female and male) or sex (women and men). Include the study's definition of sex or gender, ensuring the appropriate use of the term.

Reply: Thank you for the comment. We unify gender in the article. 

2. Were the medication use, depression duration, physical activity level, and physical modality obtained from the participants? Were the participants diagnosed with depression before the study?

Reply: Thank you for your comment. Unfortunately, we did not evaluate information regarding medication use and the duration of depression. Due to our focus on providing an association between muscle strength and depression within a large research sample, we chose to utilize a standardized questionnaire (BDI-II), which has been shown to be highly correlated with clinically diagnosed depression. Furthermore, this evaluation method is widely used in population-based studies. While information about the participant's physical activity level and modality could be informative, it was not obtained in the current study. Our primary focus was to evaluate the muscle strength profile of the knee joint using the gold standard method, which highly correlates with the amount of participants' physical activity.

Considering the population-based nature of the study, the participants had not been diagnosed with depression before their involvement in the study. 

3. Was the statistical analysis adjusted for weight?

Reply: Thank you for the question. Yes, we adjusted the muscle strength of flexors and extensors of the right and left extremities by body weight in kilograms and then standardized with z-score. Based on this we divided participants into three tertiles (T1-T3). We also added this information to the manuscript.

4. Lines 172 and 173: Did the participants with depression have a healthy weight or normal BMI?

Reply: Thank you for the recommendation. We have corrected the BMI classification and provided a better definition in the Materials and Methods section.

5. Lines 274-276. Please add a hypothesis for the association between normal BMI and depression. 

Reply: Thank you for a suggestion. We added a hypothesis that can explain the association between normal BMI and depression in our research sample.

6. Lines 277-287: Please include the meaning of the 5-HT acronym. Include the explanation related to the role of serotonin on mental health status 

Reply: Than you for the comment. We added the meaning of the acronym "5-HT" to its first mention in our manuscript. 

Thank you for your suggestion regarding the role of serotonin in mental health. We've incorporated this explanation into our manuscript to enhance its comprehensiveness.

---

## [Decision Letter · Decision Letter 1]

3 May 2024

Association between muscle strength and depression in a cohort of young adults

PONE-D-24-00268R1

Dear Dr. Vodička,

We’re pleased to inform you that your manuscript has been judged scientifically suitable for publication and will be formally accepted for publication once it meets all outstanding technical requirements.

Kind regards,

Julio Alejandro Henriques Castro da Costa

Academic Editor

PLOS ONE

Additional Editor Comments (optional):

Reviewers' comments:

Reviewer's Responses to Questions

**Comments to the Author**

1. If the authors have adequately addressed your comments raised in a previous round of review and you feel that this manuscript is now acceptable for publication, you may indicate that here to bypass the “Comments to the Author” section, enter your conflict of interest statement in the “Confidential to Editor” section, and submit your "Accept" recommendation.

Reviewer #1: All comments have been addressed

Reviewer #2: All comments have been addressed

2. Is the manuscript technically sound, and do the data support the conclusions?

Reviewer #1: Yes

Reviewer #2: Yes

3. Has the statistical analysis been performed appropriately and rigorously? 

Reviewer #1: Yes

Reviewer #2: Yes

4. Have the authors made all data underlying the findings in their manuscript fully available?

Reviewer #1: Yes

Reviewer #2: Yes

5. Is the manuscript presented in an intelligible fashion and written in standard English?

Reviewer #1: Yes

Reviewer #2: Yes

6. Review Comments to the Author

Reviewer #1: Thank you for you attention, the paper was improved

Thank you for you attention, the paper was improved

Reviewer #2: The new version of the manuscript has been improved. The authors have addressed all comments raised in the first round. My recommendation is to accept it.

7. PLOS authors have the option to publish the peer review history of their article (what does this mean?). If published, this will include your full peer review and any attached files.

Reviewer #1: No

Reviewer #2: No

---

## [Editor Report · Acceptance letter]

7 May 2024

PONE-D-24-00268R1 

PLOS ONE

Dear Dr. Vodička, 

I'm pleased to inform you that your manuscript has been deemed suitable for publication in PLOS ONE. Congratulations! Your manuscript is now being handed over to our production team.

Kind regards, 

on behalf of

Dr. Julio Alejandro Henriques Castro da Costa 

Academic Editor

PLOS ONE